# Tetratrichomoniasis in the Geese Flock—Case Report

**DOI:** 10.3390/pathogens11111219

**Published:** 2022-10-22

**Authors:** Piotr Falkowski, Andrzej Gaweł, Kamila Bobrek

**Affiliations:** Department of Epizootiology and Clinic of Bird and Exotic Animals, Wroclaw University of Environmental and Life Sciences, 50-375 Wrocław, Poland

**Keywords:** geese, tetratrichomoniasis, *Tetratrichomonas gallinarum*

## Abstract

Infections caused by tetratrichomonas are commonly observed in geese. Most cases are subclinical, and the clinical form of the disease manifests itself with a greater mortality and the presence of caseous content in ceca. We describe the case of tetratrichomoniasis in a geese flock caused by *Tetratrichomonas gallinarum*, with the genetic analysis of the isolate being based on the fragments of 18SrRNA and ITS1-5.8rRNA-ITS2.

## 1. Introduction

Flagellata, a member of the Trichomonadidae family, are the cause of an infection of the digestive tract in birds. The disease may affect the upper gastrointestinal tract, and the main agent is *Trichomonas gallinae*, or the lower intestinal tract, and the large intestine is infected by *Tetratrichomonas* spp. [1], mainly *T. gallinarum*. Trichomonads, of a small oval shape, 6.5–11.5 × 3.5–8, are micrometer parasites with four flagella on the anterior pole with well-framed axostyle on the posterior pole, showing a variety of pathogenicity within species and strains. In bird species, a *Tetratrichomonas* infection can cause high mortality rates, but in many cases it can be asymptomatic or mild, which is not observed in the clinical signs [2]. The prevalence of *Tetratrichomonas* spp. in geese flocks is high; the research showed that 88% of the geese parental flocks are positive for the presence of this pathogen [3]. Despite this fact, cases of higher mortality due to tetratrichomoniasis in geese are not common. Pecka [4] reported that an experimental geese infection caused by *T. gallinarum* culture inoculation did not cause pathological changes in the caeca despite the large number of protozoans, but lesions such as chronic necrotic ceca inflammation in geese were observed in Jugoslavia and Poland [5].

This case report describes the outbreak of tetratrichomoniasis in the parental flock of geese with the genetic analysis of the isolates.

## 2. The Case Description and Sample Collection

In a 3-year parental geese flock of three thousand birds, in October, a higher mortality rate than usually was observed. On each day, 3–4 geese were found dead, when the normal mortality rate was about 5 birds per month. The 4 dead geese were delivered for a necropsy examination to the Department of Epizootiology and Clinic of Bird and Exotic Animals, Faculty of Veterinary Medicine in Wrocław. During necropsy, all of the organs of each bird were examined. The only lesions observed in all the birds were the caseous cores in the ceca, with a brownish necrotic content (Figure 1), and perihepatitis was observed in one goose. The ceca wall and core fragments were taken for DNA isolation and for the microbiological examination, swabs from the liver, lung, heart, and ceca were taken and delivered to the Epi-Vet veterinary laboratory (Faculty of Veterinary Medicine, Wrocław)

The DNA was isolated with a commercial kit (Genomic Mini, A&A Biotechnology, Gdansk, Poland) according to the manufacturer’s instructions from the ceca wall and the caseous cores of necropsied birds. A PCR was performed with two different protocols. The first PCR was carried out with primers for the 18S rRNA fragment Tgf: 3′- GCA ATT GTT TCT CCA GAA GTG -5′ and Tgr: 3′- GAT GGC TCT CTT TGA GCT TG -5′, [6] and the second with trichomonad-specific primers TFR1(5′-TGCTTCAGTTCAGCGGGTCTTCC-3′) and TFR2 (5′-CGGTAGGTGAACCTGCCGTTGG-3′) for the fragment ITS1- 5.8S- ITS2 [7]. Both reactions were carried out using 25 µL of PCR Mix Plus (AA Biotechnology), 2 µL of 10 pM forward and reverse primers, 8 µL of template, and 13 µL of sterilized distilled water. The cycling parameters for the 18SrRNA amplification were as follows: the initial denaturation step at 95 °C for 15 min, 40 cycles with heat denaturation at 94 °C for 30 s, primer annealing 60 °C for 30 s and extension at 72 °C for 1 min, and a final extension at 72 °C for 10 min [7]. The cycling parameters for the amplification of ITS1-5.8S-ITS2 were: 94 °C for 1 min followed by 37 cycles of 94 °C for 30 s, 66 °C for 30 s, and 72 °C for 2 min, and a final extension at 72 °C for 5 min [8]. The PCR products were electrophoresed in a 2% agarose gel, stained with Midori Green, and visualized under a UV light (Quantity One Biorad), cut and cleaned using a Gel-out kit (A&A Biotechnology, Poland), and sent for sequencing with forward and reverse primers (Genomed, Warsaw, Poland). The obtained sequences were examined by BLAST searches. For the subgroup division of *T. gallinarum,* the concatenated data set, including the 18SrRNA and ITS1-5.8S-ITS2 sequences, was created manually [8,9]. The phylogenetic tree was prepared using the neighbor-joining method (1000 bootstrap) in MEGA X, and the lineage in the genus *Tetratrichomonas* described by Cepicka et al. [8] is listed on the right.

## 3. Results

The lesions observed during necropsy in the form of cores in the ceca in all geese were caused by *Tetratrichomonas,* because the microbiological examination excluded a *Salmonella* infection, which can cause similar lesions, and the PCR showed the presence of expected *Tetratrichomonas* products sized 600 bp and 300 bp (Figure 2). The sequencing confirmed that the obtained products are fragments of *Tetratrichomonas.* Due to the lack of registered drugs that could be used in tetratrichomoniasis therapy, the herbal product (Fitotril) was used according to the manufacturer’s instructions. During the tetratrichomoniasis outbreak, 108 geese died, which was almost 4% of the flock. Four samples, each from one necropsied bird, were positive. The products were cleaned and sequenced (Macrogen Europe, the Netherlands), but only two of the samples have correct results for both 18S rRNA and ITS1-5.8rRNA-ITS2 products. The received sequences were compared and, due to homology in the nucleotides, they have been deposited in the GenBank database, for the 18SrRNA fragment under the accession number ON413916.1 and for the internal transcribed spacer (ITS) fragment under number ON417741.1.

The phylogenetic tree based on the concatenated data set including the 18S rRNA and ITS1-5.8S rRNA-ITS2 fragments, according to Cepicka [8] and Chen [9], is given on Figure 3. The obtained in this investigation isolate belongs to group A, but is not identical with the isolates from subgroup A1 (isolate from swan) and A2 (isolate from turkey), and could be classified as another subgroup A.

## 4. Discussion

In geese, the presence of caseous cores in ceca suggests a *Tetratrichomonas* infection. In gallinaceous birds, the cheesy cores in ceca could also be an effect of other protozoan infections caused by species-specific Eimerias (*E. tenella, E. denoids*) or *Histomonas*, but those parasites do not infect geese. In gallinaceous birds, lesions caused by *T. gallinarum* have been reported sporadically in birds such as chukar partridges, mockingbird, Waldrapp ibis and white pelican in America, ducks in Germany, red-legged partridges in Great Britain, and layer chickens in The Netherlands [11,12,13,14,15,16,17].

The tetratrichomoniasis in geese with a high mortality, up from 7.5% to 25%, was described by Ziomko et al. [4] and the noted lesions were similar to those observed nowadays. The lower mortality rate in this case could have been an effect of the Autumn–Winter season and a lower survival of parasites not able to produce cysts or spores, compared to the outbreaks described by Ziomko, which were noted in the Spring–Summer season.

The *Tetratrichomonas* sp. prevalence in geese flocks is varied. Among the samples taken from the free-ranged geese in China, 0.7% were positive for *Tetratrichomonas*, a low number compared to our research on the prevalence of tetratrichomonas infection in the parental geese flocks in Poland (88%) [2,9]. Poland is the leader of geese production in the European Union, and China is the world biggest geese producer, but comparing the results would be hard because of some differences, including flagellata infections treatment. In China, the usage of metronidazole in the treatment is acceptable [18], while in Poland, due to an incomplete database for nitroimidazoles, the use of MTZ in poultry (and other food producing animals) is currently banned [19,20], so the level of infections could not be controlled. The intensity of poultry production and the intensive contact of birds with the flock result in the high number of infections and the outbreaks of the clinical tetratrichomoniasis in reproductive flocks of geese.

The lack of clinical forms of disease in most of the infected domestic geese may be caused the well-being of animals (regular feeding with good quality feed, immunoprophylaxis, etc). The other thesis is that among *T. gallinarum*, different types of strains exist, among which some could have a higher and some a lower pathogenicity. This situation was observed in another avian parasite *T. gallinae.* Those parasite isolates were classified based on the ITS1-5.8S rRNA-ITS2 fragment being put into two genotypes: A and B. The type B was present in all birds that showed macroscopic lesions, while type A was isolated mostly from birds without lesions [21,22]. Cepicka et al. [10] demonstrated in their research about *Tetratrichomonas* polymorphism, the phylogenetic analysis with the isolate from duck with salpingitis, but more research in the future is needed to investigate the presence of strains with different pathogenicity among *Tetratrichomonas* in poultry.

## Figures and Tables

**Figure 1 pathogens-11-01219-f001:**
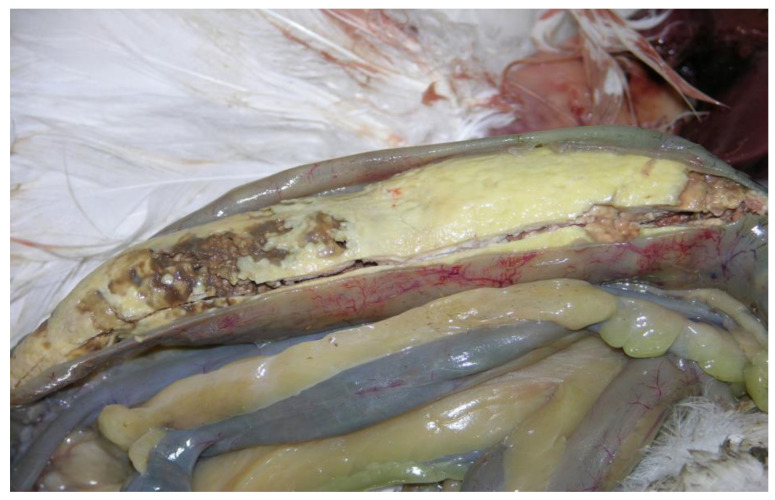
The caseous core in goose cecum with necrotic content.

**Figure 2 pathogens-11-01219-f002:**
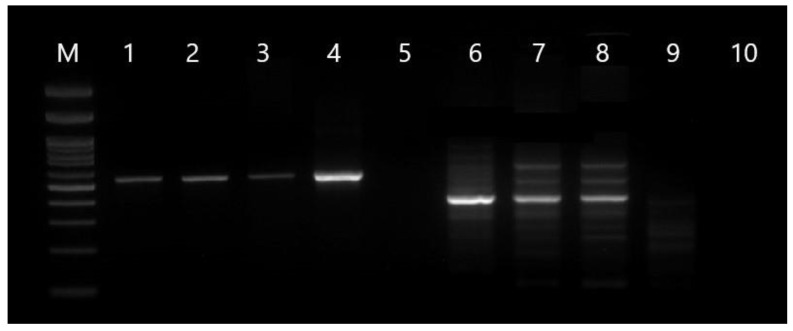
The PCR products obtained from the cecal cores of necropsied geese. Lanes 1–4, PCR products of the 18S rRNA gene. Lanes 6–9, PCR products of the ITS1–5.8S rRNA-ITS2 gene. Lane 5 and 10 were negative controls. M—Marker 100bp Plus (0.1–5 kbp) Abbexa (UK).

**Figure 3 pathogens-11-01219-f003:**
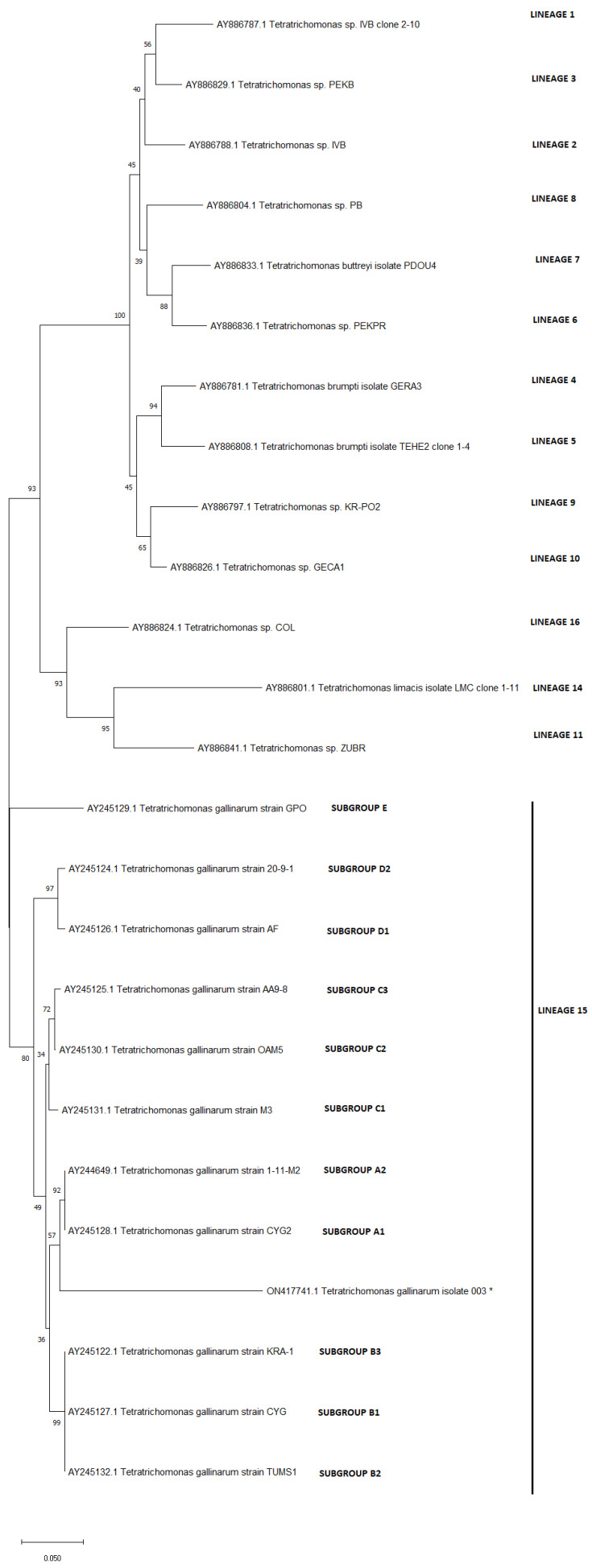
Phylogenetic interrelationships between *T. gallinarum* isolated from bird species on the concatenated ITS1-5.8rRNA-ITS2 region and 18S r RNA fragment. Isolate obtained in this study is marked with asterix. The tree was inferred by using the neighbor-joining method in MegaX with 1000 bootstrap randomization. The subgroup division came from Cepicka et al. [10].

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
