# Peer review of "Tetratrichomoniasis in the Geese Flock—Case Report"

_pathogens, 2022, doi:10.3390/pathogens11111219_

Round 1

Reviewer 1 Report

The publication is very interesting. It describes an isolate of a commonly known species of trichom which is particularly pathogenic for water birds. The described case is very carefully described and documented by genetic tests.

Author Response

Dear Editor,

we are grateful for the review and the comment that our work is very interesting.

We hope that it will go through the reviewing process sucessfully

Best regards

Reviewer 2 Report

Comments to the Author 
The manuscript ID: pathogens-1871232 reports the case of tetratrichomoniasis in a geese flock. Data presented in the manuscript is of importance for the community, since the information on tetratrichomoniasis in geese in the world is scarce. However, following adjustments should be made:
1. It is inaccurate that the author named the
Tetratrichomonas isolate in this study as  Tetratrichomonas anseris” . I think Tetratrichomonas gallinarum should be the common name for the Tetratrichomonas spp. found in domestic and wild poultry.

2. It is incorrect that the author mention that “There is no research showing the genetic relationship between genotype and pathogenicity in Tetratrichomonas spp.”in the dissicusion of the manuscript. Please see the reference by Cepicka and co-authors in 2005 and 2006, especially the lineage system proposed by the 2006 paper.

The two literatures are as follows:

Cepicka I, Kutisˇova´ K, Tachezy J, Kulda J, Flegr J (2005) Cryptic species within the Tetratrichomonas gallinarum species complex revealed by molecular polymorphism. Vet Parasitol 128:11–21

Cepicka I, Hampl V, Kulda J, Flegr J (2006) New evolutionary lineages, unexpected diversity, and host specificity in the parabasalid genus Tetratrichomonas. Mol Phylogenet Evol 39:542-551. 

3. In order to be useful for the community the phylogenetic analysis for T. gallinarum should include already published data and typing.Here I mean important work done by Cepicka and co-authors in 2005 and 2006, especially the lineage system proposed by the 2006 paper. So authors should include at least one representative of each lineage in the analysis. For clarity I would suggest the authors to label the lineages, similar as it was done for manuscript reporting the prevalence of Tetratrichomonas gallinarum and Trichomonas gallinae in three domestic free‑range poultry breeds in Anhui Province, China (Chen et al.2022)

About sequencing, the author mention that only 2 of the samples have correct results for both 18SrRNA and ITS1-5.8rRNA-ITS2 products. If the 18SrRNA and ITS1-5.8rRNA-ITS2 of the 2 samples were sequenced successfully, I would suggest the authors to renew the phylogenetic analysis using the concatenated of the 18SrRNA and ITS1-5.8rRNA-ITS2,as previously by Chen et al.2022.

4. All the Scientific and generic names in this paper should be italicized, such as

 Tetratrichomonas spp.” → " Tetratrichomonas spp.”. Please check carefully to ensure that all were corrected.

5.  2. The case description and sample collection

“PCR was performed with two different protocols. The first PCR was carried out with primers for the 18S rRNA fragment Tgf: 3′- GCA ATT GTT TCT CCA GAA GTG -5′ and Tgr: 3′- GAT GGC TCT CTT TGA GCT TG -5′, and the second with trichomonad-specific primers TFR1(5′-TGCTTCAGTTCAGCGGGTCTTCC-3′) and TFR2 (5′-CGGTAGGTGAACCT GCCGTTGG-3′) for the fragment ITS1- 5.8S- ITS2.”  Please supplied the reference that the primers come from.

6. 2. The case description and sample collection

“and sent for sequencing (Genomed, Poland). ” , Unidirectional or bidirectional sequencing ? It is very important that the results of bidirectional sequencing are more accurate than the results of Unidirectional sequencing.

7. Figure 2.  Please supplied the molecular weight and unit of  DNA marker in the figure 2.

8. Figure 3.  and Figure 4.   

(1) “18S rDNA gene ” → " 18S rRNA gene”

(2) All T. gallinarum reference sequences are labeled by T. gallinarum, their strain/isolate name, and subgroup, as previously by Chen et al.2022.

(3) Reconstruct the phylogenetic analysis using the concatenated of the 18SrRNA and ITS1-5.8rRNA-ITS2.

(4) Scientific names in Figure should be italicized

9. Discussion

(1)“ E.adenoides” → " E. adenoides

(2)“ T.gallinarum” → " T. gallinarum

(3) “ T.anseris” → " T. anseris

(4) “Trichomonas gallinae” → "T. gallinae

10. References

Please sort out the format of references according to the requirements of the magazine

Author Response

Dear Reviewer,

thank you for your review, we improved the manuscript according to your suggestions. We hope that it will find your acceptance

Best regards

Round 2

Reviewer 2 Report

Comments to the Author 
Adjustments made during the revision of the manuscript PARE-D-22-00189 need more work.

1. Different species of trichomonads were also previously classified on the basis of the hosts they infected. Now, the trichomonads found in the large intestine of gallinaceous, anseriform and other birds were all recognized as Tetratrichomonas gallinarum. It is inaccurate that the author named the Tetratrichomonas isolate in this study as  Tetratrichomonas anseris” . According to the phylogenetic interrelationships based on the concatenated ITS1-5.8rRNA-ITS2 region and 18S r RNA fragment, the T. gallinarum isolates in this study belonged to the genogroup A of T. gallinarum. And the Figure 3. and Figure 4 also suported the the T. gallinarum isolates were homologous to the isolates of other T. gallinarum isolates, not the Tetratrichomonas anseris. Please revised this in the original version of the manuscript (please see the part underlined in red annotated by reviewer in the manuscript )

2. Please delete the Figure 3. and Figure 4. The Figure 5. has been fully displayed that  the T. gallinarum isolates in this study belonged to the genogroup A of T. gallinarum, although it was not identical with isolates from subgroup A1 (isolate from swan) and A2 (isolate from turkey). If you want to keep the Figure 3. and Figure 4, please renew the Figure 3. and Figure 4, as shown in Figure 5, and made the Figure 3. and Figure 4 also supported the the T. gallinarum isolates in this study belonged to the genogroup A of T. gallinarum.

3. All the Scientific and generic names in this paper should be italicized, such as

 Tetratrichomonas spp.” → " Tetratrichomonas spp.”. Please check carefully to ensure that all (Refer to the red circle annotated by reviewer in the article, including the Scientific and generic names in the figure 3-5) were corrected.

4. Half a character between genus name abbreviation and species name

Please see the part in box annotated by reviewer in the manuscript.example: T.gallinarum” → " T. gallinarum”.

5. Use the abbreviate of the Scientific names, please see the part in triangle annotated by reviewer in the manuscript.

Author Response

Dear Editor, dear Reviewers,

Thank you for considering our manuscript for publication and providing us with your comments. Below please find the points raised by the Reviewer and our specific answers and explanations.

The green highlights in the manuscript indicate the corrected text according to Reviewers suggestions – 2nd review, yellow – 1st review.

Comments to the Author 

Adjustments made during the revision of the manuscript PARE-D-22-00189 need more work.

Comment 1.

Different species of trichomonads were also previously classified on the basis of the hosts they infected. Now, the trichomonads found in the large intestine of gallinaceous, anseriform and other birds were all recognized as Tetratrichomonas gallinarum. It is inaccurate that the author named the Tetratrichomonas isolate in this study as  “Tetratrichomonas anseris” . According to the phylogenetic interrelationships based on the concatenated ITS1-5.8rRNA-ITS2 region and 18S r RNA fragment, the T. gallinarum isolates in this study belonged to the genogroup A of T. gallinarum. And the Figure 3. and Figure 4 also suported the the T. gallinarum isolates were homologous to the isolates of other T. gallinarum isolates, not the Tetratrichomonas anseris. Please revised this in the original version of the manuscript (please see the part underlined in red annotated by reviewer in the manuscript )

Authors:

The name of the isolate in GenBank was changed. We agree that according to current knowledge it should be named T.gallinarum, and we are not able to prove that the case was caused by other that T.sallinarum species. We’re planning future investigation to prove that in birds’ species more than one Tetratrichomonas species infect the birds.

  1. Please delete the Figure 3. and Figure 4. The Figure 5. has been fully displayed that the T. gallinarum isolates in this study belonged to the genogroup A of T. gallinarum, although it was not identical with isolates from subgroup A1 (isolate from swan) and A2 (isolate from turkey). If you want to keep the Figure 3. and Figure 4, please renew the Figure 3. and Figure 4, as shown in Figure 5, and made the Figure 3. and Figure 4 also supported the the T. gallinarum isolates in this study belonged to the genogroup A of T. gallinarum.

Authors:

The mentioned figures were deleted. 

  1. All the Scientific and generic names in this paper should be italicized, such as

 “Tetratrichomonas spp.” → " Tetratrichomonas spp.”. Please check carefully to ensure that all (Refer to the red circle annotated by reviewer in the article, including the Scientific and generic names in the figure 3-5) were corrected.

Authors:

The parasite names were corrected in the text and references.

  1. Half a character between genus name abbreviation and species name

Please see the part in box annotated by reviewer in the manuscript.example: “ T.gallinarum” → " T. gallinarum”.

 Authors:

The corrections were made

  1. Use the abbreviate of the Scientific names, please see the part in triangle annotated by reviewer in the manuscript.

  Authors:

The corrections were made 

Dear Editor and Reviewers, thank you for the suggestions which improved the manuscript. We hope that the corrected text will find your approval.

Best regards

Kamila Bobrek
